# Distribution of Dissolved Nitrogen Compounds in the Water Column of a Meromictic Subarctic Lake

**Taisiya Ya. Vorobyeva** [1], **Anna A. Chupakova** [1], **Artem V. Chupakov** [1], **Svetlana A. Zabelina** [1], **Olga Y. Moreva** [1] **and Oleg S. Pokrovsky** [1,2,*]

1   N. Laverov Federal Center for Integrated Arctic Research of the Ural Branch of the Russian Academy of Sciences, 23 Severnoi Dviny Emb., 163000 Arkhangelsk, Russia; vtais@yandex.ru (T.Y.V.); anna.a.ershova@gmail.com (A.A.C.); artem.chupakov@gmail.com (A.V.C.); svetzabelina@gmail.com (S.A.Z.); mapycr1@yandex.ru (O.Y.M.)
2   Géosciences Environnement Toulouse (GET), UMR 5563, CNRS-OMP-Université Toulouse, 14 Avenue Edouard Belin, 31400 Toulouse, France
*   Correspondence: oleg.pokrovsky@get.omp.eu; Tel.: +33-5-61-33-2625

**Abstract:** In order to better understand the biogeochemical cycle of nitrogen in meromictic lakes, which can serve as a model for past aquatic environments, we measured dissolved concentrations of nitrate, nitrite, ammonium, and organic nitrogen in the deep (39 m maximal depth) subarctic Lake Svetloe (NW Russia). The lake is a rare type of freshwater meromictic water body with high concentrations of methane, ferrous iron, and manganese and low concentrations of sulfates and sulfides in the monimolimnion. In the oligotrophic mixolimnion, the concentration of mineral forms of nitrogen decreased in summer compared to winter, likely due to a phytoplankton bloom. The decomposition of the bulk of the organic matter occurs under microaerophilic/anaerobic conditions of the chemocline and is accompanied by the accumulation of nitrogen in the form of $N-NH_4$ in the monimolimnion. We revealed a strong relationship between methane and nitrogen cycles in the chemocline and monimolimnion horizons. The nitrate concentrations in Lake Svetloe varied from 9 to 13 μM throughout the water column. This fact is rare for meromictic lakes, where nitrate concentrations up to 13 μM are found in the monimolimnion zone down to the bottom layers. We hypothesize, in accord with available data for other stratified lakes that under conditions of high concentrations of manganese and ammonium at the boundary of redox conditions and below, anaerobic nitrification with the formation of nitrate occurs. Overall, most of the organic matter in Lake Svetloe undergoes biodegradation essentially under microaerophilic/anaerobic conditions of the chemocline and the monimolimnion. Consequently, the manifestation of the biogeochemical nitrogen cycle is expressed in these horizons in the most vivid and complex relationship with other cycles of elements.

**Keywords:** nitrogen cycle; ammonium; nitrate; nitrite; stratification; boreal; meromictic lake

## 1. Introduction

Nitrogen is the most important nutrient, ranking fourth among biophilic elements after carbon, oxygen, and hydrogen. The biogeochemical nitrogen cycle is one of the main cycles in the biosphere [1–6]. The nitrogen cycle is an interconnected chain of reactions for the transformation of various forms of nitrogen compounds [7], the main role of which belongs to microorganisms [8–15]. The concentration of nitrogen compounds determines the water body's biological productivity to a large extent. A change in the composition and ratio of the various forms and concentrations of nitrogen compounds indicate the direction of the dominant biological and biogeochemical processes, including the processes of self-purification of water bodies [2]. In addition to assessing water quality, information on the content and distribution of various forms of nitrogen is important for nutrient balance and the water's chemical composition [16].

Because the nitrogen cycle in aquatic systems is strongly dependent on redox conditions, chemically stratified water bodies are particularly important for the biogeochemistry of N in continental settings. Biogeochemical processes in meromictic water bodies are very interesting from the point of view of limnology and aquatic ecology because lakes of this type exhibit peculiar features: There is no water mixing between different layers and the anaerobic zone persists throughout the whole year. There are three zones in the structure of meromictic lakes: Mixolimnion—a zone in which convective and wind mixing of water occurs; monimolimnion, which is an anaerobic zone, a non-mixing layer that does not have contact with the atmosphere; and a chemocline, which is a layer of abrupt change of hydrochemical characteristics at the boundary of the mixolimnion and monimolimnion, where a complex microbial community is usually formed [17,18].

Of greatest interest in water bodies of this type are the processes occurring in the chemocline and monimolimnion of lakes. The oxygen minimum zone (OMZ), formed in the chemocline, is a site of intensive nitrogen turnover due to the influx of organic material from the overlying water layers and the formation of a large amount of organic matter in the chemocline zone. The OMZ contains unique microbial communities that use alternative electron acceptors for respiration. The OMZ conditions provide an almost complete nitrogen (N) cycle. Remineralization can occur both due to the reduction of nitrate to nitrite and can be associated with the non-assimilatory reduction of nitrate to ammonium, whereas the formation of gaseous nitrogen can occur due to heterotrophic denitrification and anammox. Many microorganisms inhabiting the oxygen minimum zones are capable of performing various functions in the nitrogen and other cycles of elements [19].

Nitrifying bacteria (both ammonium and nitrite oxidants) are present throughout the entire oxygen minimum zone. Even in waters with the lowest oxygen content, chemoautotrophic activity and oxidation of nitrogen compounds are usually detected in situ. The greatest abundance of nitrifying bacteria and their greatest activity is found in the gradient region at the upper boundary of the oxygen minimum [19,20]. A close relationship is found between nitrification and denitrification in the oxygen minimum zone where the rapid cycling of intermediate products of the nitrogen cycle, including nitrate, nitrite, and nitrogen oxide (I), occurs [20].

An increase in exports from mixolimnion and the formation of a large amount of organic matter in the chemocline (hence, respiration within this zone) leads to the depletion of dissolved oxygen, followed by other main alternative electron acceptors (nitrate, sulfate, etc.). Some of the sulfide/sulfur oxidizers can use $NO_3^-/NO_2^-$ as electron acceptors and release gaseous $N_2$ and/or $N_2O$, while the released energy is used to fix inorganic carbon via so-called chemolitoautotrophic denitrification. The latter process can also be associated with hydrogenotrophy, methanotrophy, and iron oxidation [19].

Microorganisms of the non-biogeochemical nitrogen cycle are also capable of influencing its transformation under the specific conditions of meromictic lakes. Microorganisms of the methane cycle (including both methanogenesis and methane oxidation) play an important role in the binding of global carbon and nitrogen cycles in microaerophilic and anoxic environments [21–26]. In addition to the fact that methanotrophic bacteria use methane as a source of carbon and energy, they also modify the nitrogen cycle, especially at the boundary of the aerobic–anaerobic interaction (chemocline zone) or just above or below it [27]. Methanotrophic bacteria can assimilate nitrogen as ammonia or nitrate and can compete with nitrifying bacteria for ammonia and oxygen [26,28,29].

Studies of small Red/Ox-stratified boreal lakes [24] have demonstrated that methanotrophs of boreal lakes can associate methane oxidation with $NO_x^-$ reduction under hypoxic conditions. Recent studies of a shallow, seasonally stratified subalpine lake [30] have shown that $NH_4^+$, being the main nutrient in this aquatic ecosystem, has a positive correlation with methane-oxidizing bacteria (MOB). The latter can metabolize $NH_4^+$ for growth. Investigations of the oxygen-stratified Lake Fohnsee (southern Germany) have shown that anaerobic methane oxidation, denitrification, and anammox can simultaneously

occur in the anoxic water column [31]. Analysis of gas taken from various anaerobic environments, where a significant amount of organic matter accumulates and decomposes with the release of methane, demonstrated that free nitrogen of biological origin can constitute up to 30% of all the released gas [7]. A relationship between methane oxidation and nitrate reduction during hypoxia has recently been shown for the Gammaproteobacterium *Methylomonas denitrificans*, attributing to this species a previously overlooked role in coupling carbon and nitrogen cycles [32].

The methane cycle in Lake Svetloe is fairly well known [33–38]. The microbial communities participating in the methane cycle are represented by several genera of methanogens and methanotrophs typical of freshwater lakes [35]. Methane oxidation was found in both oxygen and anoxic conditions with a maximum in chemocline. Methanotrophic bacteria *Methylobacter* sp. and methylotrophic *Methylotenera* sp. and *Methylophilus* sp. showed similar profiles of relative abundance throughout the epilimnion, chemocline, and hypolimnion of Lake Svetloe [35,36]. However, in contrast to fairly good knowledge of the carbon and methane cycle in Lake Svetloe, the knowledge of the nitrogen cycle remains highly limited.

This work presents the first data on the spatiotemporal dynamics of the concentrations of organic and inorganic nitrogen forms in the subarctic meromictic Lake Svetloe (NW Russia). The lake is a rare type of freshwater meromictic water body with high concentrations of dissolved methane and ferrous iron, manganese, and low concentrations of sulfates and sulfides in the hypolimnion [34,36,39,40]. Iron-rich and sulfur-depleted meromictic lakes with conditions suitable for photoferrotrophy are considered modern analogues of the ancient Archean Ocean [21,41]. The high transparency of the waters, low water color, and the distribution of the photic layer throughout the depth create favorable conditions for the development of phytoplankton and the specific microbial community of the chemocline [33,36]. Our specific objectives were to (*i*) assess vertical and seasonal dynamics of dissolved nitrogen forms and (*ii*) relate the spatial pattern of these concentrations to already known cycles of carbon, oxygen, methane, iron, and manganese in the lake, taking into account the microbiological control on the N cycle.

## 2. Study Site and Methods

### 2.1. Site Description

Lake Svetloe is located in the northern part of the boreal zone of European Russia (N 65°04.98′, E 41°06.26′), 65 km NNE of Arkhangelsk, and its watershed is not subjected to any direct anthropogenic influence. The maximum depth of the lake is 39 m (Figure 1).

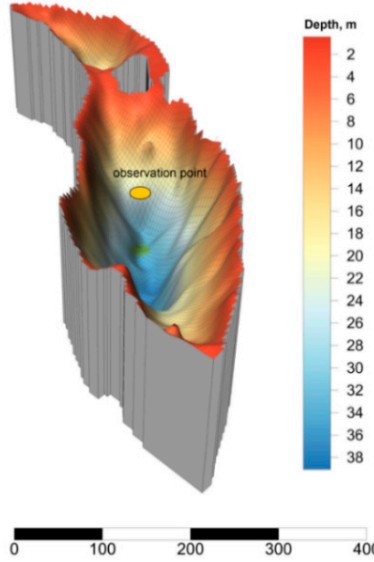

**Figure 1.** 3D model and bathymetric scheme of Lake Svetloe (scale bar in meters).

Two pronounced relief depressions conditionally divide the lake into two parts with maximum depths of 39 and 20 m (Figure 1). In accordance with the meromictic status, three layers in the water column of the lake can be distinguished: (1) Mixolimnion (from 0 to 20 m), subject to convective mixing throughout the year and exhibiting aerobic conditions; (2) chemocline, which is a transitional zone at a depth of 20–24 m, where microaerophilic conditions are formed; in this zone, oxygen is produced by cyanobacteria and is actively consumed by other microorganisms [36]; and (3) monimolimnion (from 25 m to the bottom), which is an anaerobic layer [36,39]. Lake Svetloe is characterized by a predominance of autochthonous dissolved organic matter with a low content of dissolved organic carbon (83.3 to 333.3 $\mu M$, [39,40]) and high water transparency (12 to 16 m Secchi depth).

The water in the chemocline of Lake Svetloe has a faint pink color of varying intensity, due to the development of phototrophic communities inhabiting the boundary of the aerobic and anaerobic zones. Studies of phototrophic bacteria in the communities of the chemocline zone have demonstrated that the dominant bacteria are oxygenic phototrophic cyanobacteria of the genus Synechococcus (maximum development at a depth of 23 m) [33,36], which have positive chemotaxis to nitrogen sources [42–44] and are capable of fixing molecular nitrogen [45].

### 2.2. Sampling and Analyses

Sampling was carried out from 2010 to 2016 and included 30 survey campaigns during all hydrological seasons. Sampling was carried out over the entire water column from the surface to the bottom, with a step of 1 to 6 m. Water samples were taken at the deepest point of the lake approximately in the middle of the water body (N 65°4.975′, E 41°6.497′) from a PVC boat from May to October, and from the ice in winter (November to April) using a pre-cleaned polycarbonate horizontal water sampler (Aquatic Research Co, ID, New York, NY, USA).

A water sample for measurements of nitrite, nitrate, and total nitrogen was collected with a water sampler, to the tap of which a PVC outlet tube was attached. First, the bottle was rinsed 2–3 times with water from the sampler. Then the outlet tube of the sampler was lowered to the bottom of the bottle and began to fill it with water, passing several volumes of water, that is, until the water that was in contact with the air in the bottle was entirely displaced. As such, most of the water sample was not in direct contact with the atmosphere. Then the polypropylene containers were closed with screw caps with an inner cone with slight compression of the bottle itself. This technique minimized the possibility of having residual air in the vial. It should be noted that when examining the water samples in the laboratory after transportation, we did not observe the precipitation of Fe (III) hydroxide, which would inevitably form during the oxidation of Fe (II) in a neutral to slightly alkaline medium. Chemical analysis of water samples was carried out on the day of sampling.

A water sample for the determination of $N-NH_4$ was taken in a separate glass container in the same way as for other forms of nitrogen described above. Fixation of $N-NH_4$ with reagents was carried out immediately after filling the bottle with water. In addition, at the beginning of the study of Lake Svetloe, we carried out a parallel determination of the content of $N-NO_2$ and $N-NO_3$ directly on the lake shore, and afterwards, the samples were delivered to the laboratory. The results of parallel measurements were reproducible within the uncertainty of analyses.

The nitrogen forms' ($N-NO_2$, $N-NO_3$, $N-NH_4$, TN) determinations were based on colorimetric assays [40,46]. The indophenol blue method was used to measure ammonium ($N-NH_4$), with a relative error of up to 12% and a detection limit of 1 $\mu M$. A spectrophotometric method employing sulfanilamide and N-(1-naphthyl)ethylenediamine dihydrochloride was used for the analysis of nitrite ($N-NO_2$), with a relative error of up to 18% and a detection limit of 0.03 $\mu M$[47,48]. A spectrophotometric method employing sodium salicylate was chosen for the determination of nitrate ($N-NO_3$), with a relative error of up to 18% and a detection limit of 7 $\mu M$. The total dissolved organic nitrogen (Norg) was evaluated from the difference between the total dissolved nitrogen (TN, persulfate oxidation, relative

error up to 12%, detection limit of 18 μM) and the total dissolved inorganic nitrogen (DIN, or the sum of nitrite, nitrate, and ammonium nitrogen). All concentrations for the nitrogen compounds are given in micro mole per liter (μM) of nitrogen. Generally, we obtained a single result for each sample from a discrete horizon. The sample size and reliability of the mean estimates were achieved by the number of vertical surveys along the water column in different seasons (30 profiles).

## 3. Results

### 3.1. Main Hydrochemical Characteristics

The main hydrological parameters of the lake are presented elsewhere [34,39,40]. The vertical profile of oxygen, temperature, conductivity, and water density are shown in Figure S1 of the Supplementary Material. The numerical values of physicochemical parameters (oxygen, temperature, and pH) of the water column of Lake Svetloe in different seasons of the year are shown in Table S1 of the Supplementary Material. Seasonal water temperature variation occurs to a depth of 22 m; further below, the water temperature decreases with a gradient of about 0.7 °C/m, and below 27 m, the water temperature is in the range of 3.5–3.6 °C all year round. Specific conductivity of the surface horizons ranges from 150 to 250 μS/cm, whereas in the bottom horizons, it ranges from 340 to 380 μS/cm. In the mixolimnion, the pH varies from 7 to 8.4 (maximum values during periods of mass phytoplankton bloom), in the chemocline, the pH varies from 6.8 to 8, and in the monimolimnion, a narrower range of variation is observed (from 6.8 to 7.6). A general decrease in pH is observed with depth. At depths of 24–25 m, there is a slight increase in pH values associated with the development of anoxygenic bacteria or cyanobacteria.

The dissolved oxygen concentration in the summer period in the thermocline (3–7 m) reaches its maximum values (over 375 μM). In the lower layers, it gradually decreases to the minimum values in the chemocline. In winter, high oxygen content is noted throughout the mixolimnion with an abrupt increase above the chemocline. In the chemocline, at depths of 21 to 24 m, a layer with microaerophilic conditions is formed, which persists throughout the year. The absence of oxygen from a depth of 24–25 m down to the bottom was noted. In the chemocline layers, the sign of the redox potential changes. At the depth of 22–24 m, according to estimates (our data and [36]), Eh has a negative value, $-107$–$(-168)$ mV.

The lake is a meromictic waterbody of iron-manganese type. The sulfide concentration is lower than that of iron and manganese, which is quite rare among meromictic lakes [39]. The most characteristic intervals (25–75%) of the concentrations of iron and manganese (dissolved fraction <0.45 μm) in the zones of mixolimnion, chemocline, and monimolimnion are equal, respectively: Fe—0.01–0.028, 0.6–41.7, 111–144 μM; Mn—0.006–0.028, 12–43, 52–54 μM. A sharp increase in the concentration of Fe and Mn also occurs when the Red/Ox conditions change.

Lake Svetloe exhibits a low concentration of dissolved organic carbon (DOC) (the minimum and maximum values were 68 μM and 358 μM, respectively); the average value over the entire water column was $167 \pm 64$ μM. The maximum values of the DOC concentration in the mixolimnion were observed in the surface layers during the phytoplankton bloom. The minimum and maximum values of dissolved inorganic carbon (DIC) were 1725 μM and 5704 μM, respectively; the average value over the entire water column was $3407 \pm 130$ μM.

### 3.2. Nitrogen Compounds

The various forms of nitrogen concentration of the water column of Lake Svetloe in different seasons of the year are shown in Table S1 of the Supplementary Material. The distribution of the $N-NH_4$ concentration in the water column of Lake Svetloe is characterized by a sharp increase in the chemocline during all seasons (Figures 2a and 3). The maximal concentration reaches, on average, 144 to 177 μM in the bottom layer. In the mixolimnion, $N-NH_4$ concentrations rarely exceeded 1.2 μM. In summer, the concentrations of $N-NH_4$ in the surface layers were lower (up to 0.7 μM) than in winter.

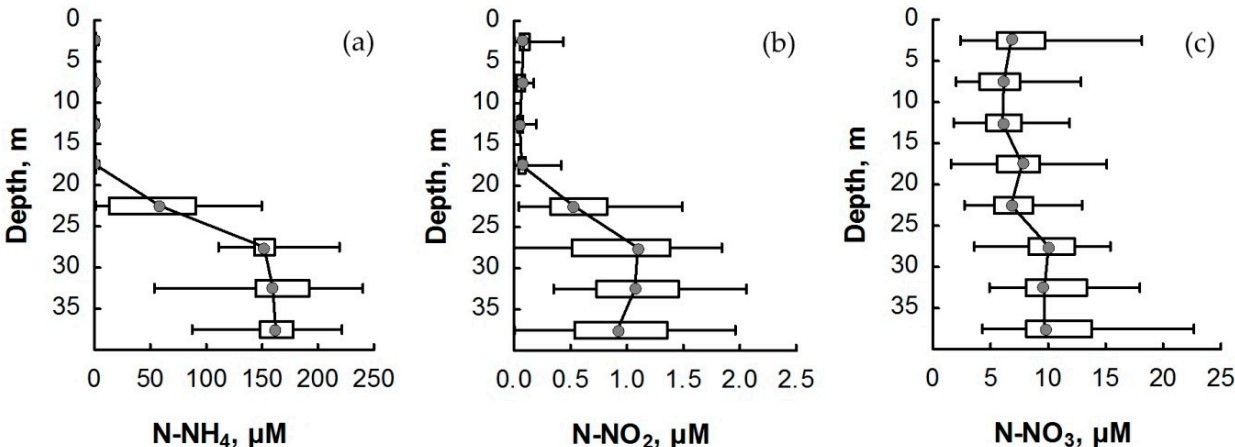

**Figure 2.** Distribution of (**a**) ammonium, (**b**) nitrite nitrogen, and (**c**) nitrate nitrogen in the water column of Lake Svetloe (in μM) averaged over four seasons from 2010 to 2016. Middle point—median; box value—percentiles (25–75%); whisker value—min–max.

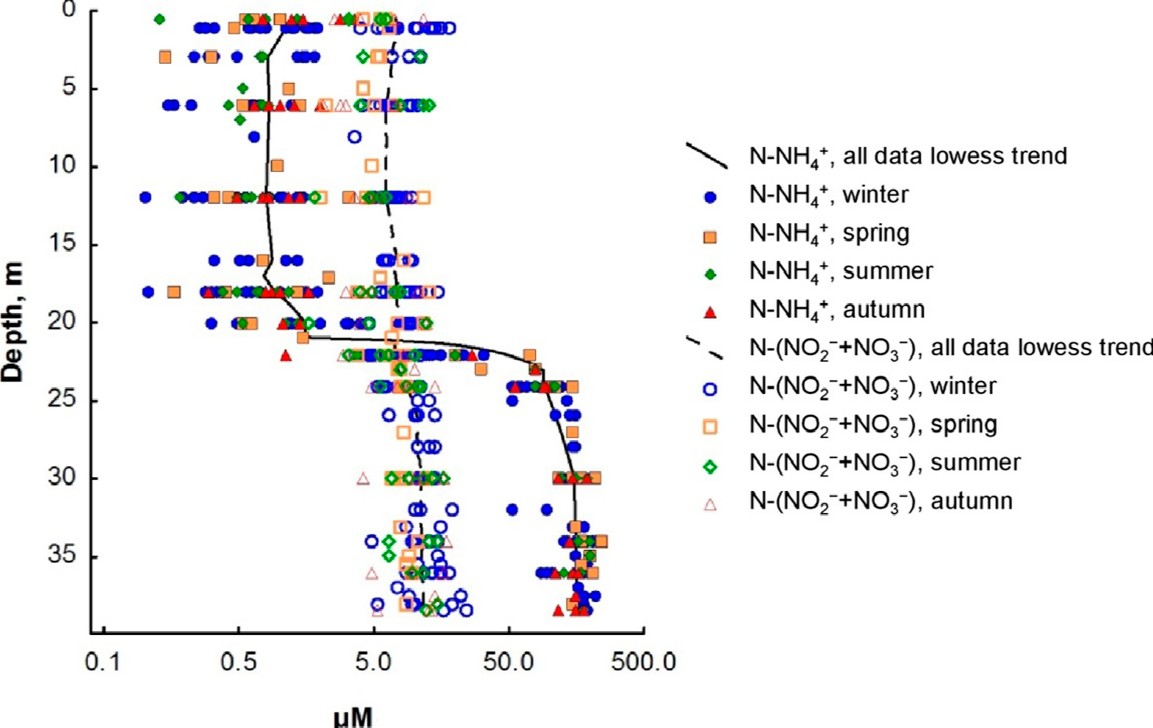

**Figure 3.** Seasonal dynamics of ammonium and the amount of nitrite and nitrate nitrogen in the water column of Lake Svetloe (in μM). Winter (November–April), spring (May), summer (June–August), autumn (September–October).

The distribution of $N-NO_3$ during all seasons demonstrated a slight decrease in concentration in the chemocline zone with a subsequent increase in the deeper layers (Figures 2c and 3). The concentration of $N-NO_3$ in the monimolimnion did not exceed the values achieved in the mixolimnion and varied in the range of 9–13 μM. The TN concentration in the mixolimnion varied within the range of 9 to 23 μM (Figure 4).

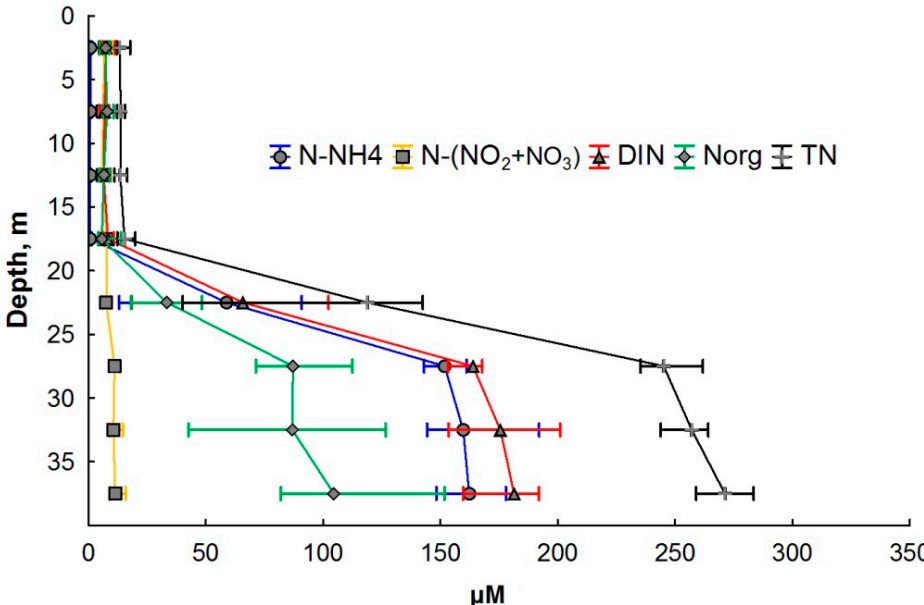

**Figure 4.** Various nitrogen forms' distribution in the water column of Lake Svetloe (in μM). Middle point—median; whisker value—percentiles (25–75%).

There was a sharp increase in TN in the chemocline (by one order of magnitude). The TN concentration reached 293 μM near the bottom layer. The share of Norg in the mixolimnion varied from 16 to 75% of TN, and in the monimolimnion, it ranged from 12 to 95% of TN (Figure 4).

The vertical distribution of mineral forms of nitrogen (Figure 3) demonstrated that the N-NO$_3$ form always predominated in the mixolimnion of Lake Svetloe. The predominance of N-NH$_4$ is observed from a depth of 22 m; the content of N-NO$_2$ in all seasons was very low and varied within the range of 0–1.4 μM, which is 0.1–4% of the sum of inorganic forms of nitrogen (Figure 2b).

## 4. Discussion

Comparative analysis of inorganic forms of nitrogen concentration in some meromictic lakes (Table 1) showed that there is a similar tendency of an increase in the content of ammonium from the mixolimnion to the monimolimnion with a sharp increase in the N-NH$_4$ concentration in the chemocline. In the vertical distribution of nitrate concentration, there are similar features between lakes Svetloe (this study) and Pavin [49]. In both lakes, a decrease in the concentration of nitrate nitrogen in the chemocline zone is observed. In contrast to the compared meromictic lakes, the concentration of nitrates in the monimolimnion in Lake Svetloe remains at the level of the mixolimnion. A similar dynamic is observed only in the meromictic Lake Zug, where nitrate is present in the anoxic hypolimnion [50].

The surface horizon of Lake Svetloe (0.5–3 m) is almost always distinguished by an increased number of heterotrophic bacteria, including ammonifying ones, and a slight excess of ammonium concentrations in comparison with the lower layers of the epilimnion, which may indicate a more active ammonification. By the content of nitrogen compounds, the mixolimnion layer can be characterized as oligotrophic, which is also confirmed by the values of the isotopic composition of suspended organic carbon: For the upper oxygen layer, the values $\delta^{13}C_{org}$ of −27.2 to −30.2‰ are common for the phytoplankton of oligotrophic freshwater lakes [36].

**Table 1.** The content of mineral forms of nitrogen in Lake Svetloe (25–75%) and other meromictic lakes in the world. Concentrations are given in μM.

| Lake | N-NH$_4$ | | | N-NO$_2$ | | | N-NO$_3$ | | | Reference |
|------|-----|------|-------|-----|------|-------|-----|------|-------|-----------|
| | Mix | Chem | Monim | Mix | Chem | Monim | Mix | Chem | Monim | |
| Lake Svetloe | 0.5–1.2 | 13–91 | 144–177 | 0–0.07 | 0.3–0.8 | 0.6–1.4 | 5–9 | 5–8 | 9–13 | This Research |
| Lake Matano | up to 20 | 20–300 | 200–300 | <0.1 | | | all layers < 0.1 only 90–100 m 0.1–0.2 | | | [21] |
| Lake Pavin | – | – | max 389 | – | – | – | max 32 | – | – | [49] |
| Lake La Cruz | 1.1–12.8 | 122.2 | up to 3000 | 0.04–0.11 | 0 | 0 | 5–20 | 0 | 0 | [18,51] |
| Kabuno Bay | – | – | – | up to 1 | N-(NO$_2$ + NO$_3$) Up to 0.5 in dry season up to 1.5 | | up to 0.5 | | | [52] |
| Lake Kuznechikha | 1.4–3.5 | 17.8–50 | 83–121 | – | – | – | – | – | – | [53] |
| Char lake | – | – | – | – | – | – | <0.7 | – | <0.7 | [54] |
| Lake Chernoe | – | – | up to 265 | – | – | – | – | – | – | [55] |
| Hall Lake | 0 | – | 305 | – | – | – | 13.3 | – | 4.4 | [18,56] |
| Lake Zug | | 10 | | – | – | – | 20–25 | 17–18 | 1–5 | [50] |

The microbial communities of the mixolimnion of Lake Svetloe are typical of those for aerobic lake environments [57] and are mainly represented by microbial groups participating in the initial stages of organic matter decomposition [35,36]. Measurements of the rate of dark CO$_2$ assimilation indicate low activity of heterotrophic bacterioplankton [35,36].

Aerobic nitrification of ammonia to nitrite and nitrate is a major process in the global nitrogen cycle. Low concentrations of N-NH$_4$ and high concentrations of nitrates were observed in the mixolimnion, except for the peak periods of phytoplankton development, when the concentration of nitrates can decrease below 1.6 μM. This testifies active processes of nitrification, which proceed to the final compound of nitrate in the water of Lake Svetloe. Low N-NO$_2$ concentrations in the mixolimnion and a large amount of N-NO$_3$ in this layer may indicate that nitrification processes proceed to the final stage in the presence of a sufficient amount of oxygen in the mixolimnion.

The peak in the development of phytoplankton communities (maximum oxygenic phototrophic activity [35,36]) is most often observed at a depth of 6–10 m, which is also evidenced by an increase in pH and dissolved oxygen concentration at this depth. There is also a slight increase in the concentration of organic nitrogen and a decrease in the inorganic nitrogen concentration, notably of nitrates.

The concentration of dissolved oxygen already starts to decrease at 16–17 m, reaching the minimum at 20–21 m. As a result, the intensity of aerobic processes sharply decreases and organic matter that is not completely mineralized enters the chemocline. In the mixolimnion of the lake, the suspended organic matter content and the total number of microorganisms are significantly lower than in the chemocline and monimolimnion [35,36].

The largest amount of organic matter is formed in the chemocline zones. The major peak of microbial processes in the chemocline coincides with the local maximum for carbon isotope fractionation [35,36]. Oxygenic phototrophic cyanobacteria, highly abundant in the chemocline, provide specific conditions for the functioning of the ecosystem. For example, the first peak of oxygenic photosynthesis is observed in the mixolimnion of the lake, and the second peak of oxygenic and anoxygenic photosynthesis is observed in the chemocline [35,36]. In accordance with previous works [58], we believe that the presence of constant stratification in such water bodies creates a "trap" for nutrients in the anaerobic zone. The organic matter formed in the chemocline enters the anaerobic layers, monimolimnion, where the anaerobic ammonification takes place.

Given that the majority of organic matter in Lake Svetloe undergoes biodegradation essentially under microaerophilic/anaerobic conditions of the chemocline and monimolimnion, the manifestation of the biogeochemical nitrogen cycle is expressed in these horizons in the most vivid and complex relationship with other cycles of elements. Due to the production of a significant amount of organic matter in the chemocline [36] and additional accumulation of OM in the chemocline linked to a sharp increase in the ver-

tical density gradient [18], a sharp increase in ammonium ions (up to 200 times) concentration occurs within this relatively thin water layer. A sharp increase in the concentration of ammonium ions in the chemocline of the lake in comparison with the mixolimnion indicates intensive ammonification processes. As a result, at the boundary of aerophilic/microaerophilic/anaerobic conditions, aerobic and facultative anaerobic ammonifiers are likely to develop intensively.

The chemocline zone can be considered a very strong biological filter of methane. Indeed, in this zone, the proliferating methanotrophic bacteria [35,36,59] use nitrogen in the form of ammonia/ammonium or nitrate in the process of methane oxidation. Methanotrophic bacteria can assimilate nitrogen as ammonia or nitrate and can compete with nitrifying bacteria for ammonia and oxygen [28]. It is possible that the nitrifying bacteria development in the chemocline of Lake Svetloe is limited due to the active process of methane oxidation since simultaneous nitrification and methane oxidation were observed only while maintaining relatively high levels of ammonium and oxygen [27]. The use of nitrate and/or suppression of the nitrification process is consistent with a slight decrease in nitrate concentrations in the chemocline (Figure 2).

The oxidation of ammonia is carried out by both bacteria and archaea Crenarchaeota or Thaumarchaeota [60,61], which can play a dominant role in the oxidation of ammonia. The distribution of ammonium-oxidizing bacteria (AOB) and ammonium-oxidizing archaea (AOA) along the water column and during the seasons is influenced by environmental factors, such as oxygen concentration [62,63], ammonia concentration [64,65], pH, and concentrations of nitrites, nitrates, and phosphates [66].

Under conditions of low oxygen concentration, ammonium-oxidizing archaea are highly stable [67–69]. In addition, they are probably not influenced by methanotrophic bacteria, as in AOB. The presence of archaea Thaumarchaeota in the oxygen minimum zone [35] at a depth of 17–22 m is consistent with sufficient ammonium concentration and the presence of oxygen. As is known, under anaerobic conditions, ammonification is slow and does not lead to significant ammonia/ammonium production. Therefore, a sharp increase in the concentration of ammonium in the monimolimnion is not observed.

Gamma, δ-, and ε-proteobacteria have an ammonifying capacity, and some of them are capable of other alternative respiratory pathways, such as dissimilatory iron, sulfate, and sulfur reduction [70]. The ability to ammonify nitrates and nitrites has been also found in chemolithotrophs such as oxidizers of Fe (II), hydrogen and sulfides, as well as anammox bacteria [70,71]. Overall, the effect of high concentrations of ammonium on a decrease in the assimilability of nitrates in bacteria, cyanobacteria, and archaea, including halophilic species, was found. When microorganisms capable of assimilating nitrate are exposed to ammonium, the cells' ability to use nitrate is drastically reduced [70,72–75].

There is another pathway for the formation of ammonium, the so-called dissimilatory nitrate reduction to ammonium (DNRA) process, also known as nitrate/nitrite ammonification [19,70]. Under microaerophilic or anaerobic conditions, DNRA may occur with the participation of bacteria, cyanobacteria, and archaea, including some halophilic species [70,72–75]. For this, nitrate is transported into cells by an active transport system and is reduced to ammonium by the sequential action of assimilatory nitrates (Nas) and nitrite reductases (NiR) [73,76,77]. In natural environments, dissimilated nitrate reduction occurs under a high ratio of ammonium nitrogen to nitrate nitrogen [7]. In Lake Svetloe, the ratio of ammonium to nitrate strongly increases from a depth of 24 m (Figure 5), which may indicate the predominance of dissimilatory nitrate reduction over denitrification under anaerobic conditions.

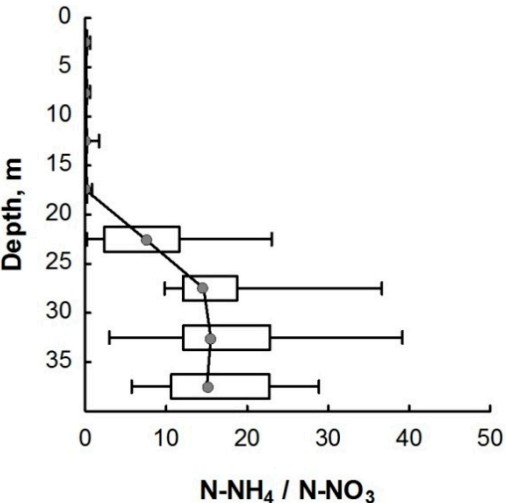

**Figure 5.** The ratio of the concentrations of ammonium and nitrate nitrogen in the water column of Lake Svetloe. Middle point—median; box value—percentiles (25–75%); whisker value—min–max.

The monimolimnion of Lake Svetloe remains a fairly stable environment. The decomposition of the main fraction of organic matter, which occurs under the anaerobic conditions of the monimolimnion, is accompanied by the accumulation of nitrogen in the N-NH$_4$ form, as evidenced by high concentrations of N-NH$_4$ in all layers regardless of the season. In the monimolimnion of lakes, nitrate is usually present in minimal concentrations, but for Lake Svetloe, rather high concentrations are observed up to the bottom sediments (Figure 2).

Nitrification usually does not occur under anaerobic conditions, whereas the denitrification processes are quite pronounced. Most likely, oxygen is still present in the chemocline and monimolimnion of Lake Svetloe. Microbiological studies of Lake Svetloe [35–37] reported a high relative abundance of cyanobacteria in the chemocline zone. Cyanobacteria were also found in the monimolimnion down to the bottom sediments, but in a lower relative abundance. Oxygen produced by photosynthetic cyanobacteria can be used by nitrifying bacteria, ammonium-oxidizing bacteria, or ammonium-oxidizing archaea, bacteria of the Nitrospirae family, found in small amounts in the chemocline of Lake Svetloe [35], representatives of which carry out the complete oxidation of ammonium into nitrate (comammox).

Comammox was discovered and described relatively recently [78,79]. After oxygen, the nitrate ion is the next preferred electron acceptor for respiration [80]. Moreover, it is the nitrate form of nitrogen that is the first preferred alternative electron acceptor among all nitrogen compounds. Consequently, nitrate ions are primarily subjected to assimilation/dissimilation under anaerobic conditions. The presence of nitrates down to the bottom horizons may indicate, as mentioned above, the presence of oxygen, which promotes the formation of nitrates.

It is also possible that under conditions of high concentrations of manganese and ammonium, anaerobic conditions occur with the formation of nitrates according to the reaction [81,82]:

$$4\,MnO_2 + NH_4{}^+ + 6\,H^+ \Rightarrow 4\,Mn^{2+} + NO_3{}^- + 5\,H_2O$$

Usually, the oxidation of Mn to form Mn particles occurs where Mn (II) and oxygen are present. Manganese oxides fall below the redox boundary, where there are very few sulfides acting as a reducing agent, but a sufficient amount of ammonium [50,83]. Manganese reduction can occur under oxidized conditions [84,85]; therefore, oxidation and reduction can occur simultaneously in the same water layer. For meromictic Lake Zug, it has been suggested that ammonium oxidation in the presence of manganese can be a source of nitrates, thus increasing their concentration up to 18 μM in the monimolimnion [50]. It

is important to note that in the anaerobic conditions in the monimolimnion layer of Svetloe Lake, nitrate concentration is still maintained at the 10 μM level, which is even higher than surface concentration. Considering the high concentration of iron and manganese, and in agreement with available literature information, it is most possible that redox-sensitive metals support ammonia oxidation to nitrite and nitrate.

However, we could not unambiguously identify the driver of manganese oxidation in the anaerobic horizons of the Lake Svetloe water column. It should be noted that a measured concentration value of zero does not mean the complete absence of a component. Oxygen can be found in concentrations below the detection limit of the method, for example, in a form not measurable by this method (e.g., peroxide or free radical). There may be an established balance of redox processes as a result of which the formed oxygen is immediately consumed in the reaction. As such, the scenario described above is quite possible.

Microorganisms associated with the process of nitrate/nitrite-dependent anaerobic oxidation of methane (ANME-2d) [35,37], that is, archaea of ANME 2d clusters, including '*Ca*N*didatus* Methanoperedens nitroreducens' and bacteria of the NC10 type ('*Ca*N*didatus* Methylomirabilis oxyfera'), are absent or found in very small numbers in Lake Svetloe. Apparently, nitrate/nitrite-dependent anaerobic oxidation of methane is minimal [21,25].

A slight increase in nitrite in the bottom layers can occur due to the oxidation of ammonia to nitrite. In the 27–35 m horizons of Lake Svetloe, ammonia-oxidizing archaea Thaumarchaeota of the genus N*itrosopumilus* are abundant, constituting up to 6.6% of all 16S rRNA gene sequences [35,37]. Ammonia-oxidizing archaea are ubiquitous in marine, freshwater, and terrestrial ecosystems. They are now considered to be a significant contributor to the carbon and nitrogen cycle. Representatives of this genus are autotrophs that receive energy from the aerobic oxidation of ammonia to nitrite and the fixation of inorganic carbon [66,86,87].

The permanent redoxcline and simultaneous presence of ammonia and nitrite pro-vide stable conditions for anaerobic ammonia oxidation ("anammox") [88]. Anammox is the anaerobic oxidation of ammonium with nitrite, resulting in the formation of molecular nitrogen $N_2$. Although anammox is an anaerobic process, oxygen does not appear to completely suppress it up to a concentration of ~13.5 μM [89]. Anammox-conducting bacteria live in many natural ecosystems usually under conditions of limited ammonia and make a significant contribution to the nitrogen cycle [71]. The anammox reaction uses $NH_4^+$ и$NO_2^-$ in a stoichiometric ratio of about 1:1 [19]. In lake ecosystems, this process has been suggested for sediments [90] or for the chemocline of several meromictic lakes: Lake Tanganyika [91], Lake Rassnitz (Germany; [92]), and Lake Lugano (Switzerland, Italy; [93]). However, for Lake Svetloe, the anammox process is the least likely, since the bacteria Planctomycetes that carry out this process [71,94] have not been found in the lake monimolimnion [35]. It is possible that high ammonium content and low nitrite concentrations in the chemocline and monimolimnion are limiting factors for their development [19,71]. It is also not excluded that, in Lake Svetloe, anammox is additionally inhibited by methanol [89], formed during methane oxidation [95].

## 5. Conclusions

Analysis of the data on the spatio-temporal dynamics of the nitrogen forms and concentrations in the low-sulphide, ferro-manganese meromictic subarctic lake demonstrated the predominance of anaerobic processes in the nitrogen cycle. When the Red/Ox conditions change, the dominant form of mineral nitrogen changes from oxidized ($NO_3$) to reduced ($NH_4$). The N-$NO_3$ form always predominated in the mixolimnion of Lake Svetloe, and seasonal fluctuations in the nitrogen concentration were associated with phytoplankton consumption. Below the depth of 22 m, N-$NH_4$ predominated, whereas the concentration of N-$NO_2$ never exceeded 4% of the sum of inorganic forms of nitrogen. In the horizons of the chemocline and monimolimnion, there was a close relationship between the biogeochemical cycles of methane and nitrogen. This is indirectly confirmed by the

local minimum of the nitrate concentration in the chemocline, where the methanotrophic bacteria could use N-NH$_4$ or N-NO$_3$ for methane oxidation.

A peculiar feature of the nitrogen cycle in Lake Svetloe is high concentrations of nitrate in the anaerobic waters of monimolimnion. This phenomenon is possibly a consequence of the iron-manganese type of meromixia of the waterbody. Under conditions of high concentrations of manganese and ammonium, anaerobic oxidation of ammonium with the formation of nitrates could occur, where MnO$_2$ acts as an oxidizing agent. This process is described in the literature for other meromictic bodies. The functioning of the nitrogen cycle in the meromictic Lake Svetloe reflects the climatic features of the subarctic (long glacial season), geochemical factors of the catchment (low DOC content and high transparency), the type of meromixia of the lake (high concentrations of Fe and Mn), and the features of the carbon cycle (methanogenesis/methane oxidation). Further, a more comprehensive analysis of these relationships should provide the necessary information on the functioning of the ecosystem, and the biogeochemical nitrogen cycle provides such an opportunity. Further work is required in this direction, in particular, the assessment of the content of gas components of the nitrogen cycle such as N$_2$ and N$_2$O. In this regard, NO might be especially important among other processes as it is closely related to denitrification.

**Supplementary Materials:** The following are available online at https://www.mdpi.com/article/10.3390/nitrogen2040029/s1, Table S1: The various forms of nitrogen concentration and other physicochemical characteristics of the water column of Lake Svetloe in different seasons of the year. All concentrations are given in μM (for the nitrogen compounds μM nitrogen in L), Figure S1: Vertical and seasonal variability of values of temperature (A), oxygen concentration (B), specific electrical conductivity (C) and density (D).

**Author Contributions:** Conceptualization, T.Y.V. and O.S.P.; methodology A.A.C.; software, A.V.C.; validation, O.S.P. and S.A.Z.; formal analysis, A.A.C.; investigation, A.A.C., A.V.C. and O.Y.M.; writing—original draft preparation, T.Y.V., A.A.C. and A.V.C.; writing—review and editing, O.S.P. and S.A.Z.; supervision, O.S.P.; project administration, T.Y.V.; funding acquisition, T.Y.V. All authors have read and agreed to the published version of the manuscript.

**Funding:** This research was funded by the State Task AAAA-A18-118012390200-5.

**Institutional Review Board Statement:** Not applicable.

**Informed Consent Statement:** Not applicable.

**Data Availability Statement:** The data presented in this study are available in the Supplementary Materials.

**Acknowledgments:** We are grateful to Klimov S.I. for the hydrological studies of Lake Svetloe.

**Conflicts of Interest:** The authors declare no conflict of interest.

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
