# Peer review of "Distribution of Dissolved Nitrogen Compounds in the Water Column of a Meromictic Subarctic Lake"

_nitrogen, doi:10.3390/nitrogen2040029_

Round 1
Reviewer 1 Report
- The most interesting finding in this paper is that in the anaerobic condition in the monimolimnion layer, nitrate concentration still maintains at 10 μM level which is even higher than surface concentration. Considering the high concentration of iron and manganese, it is most possible that redox-sensitive metals support ammonia oxidation to nitrite and nitrate. I suggest that the article need more discussions about iron and ammonium oxidation under anaerobic conditions. Further research about the relationship between metals and kinetic experiments of nitrification is suggested to be carried out in the future. On the other hand, why nitrite and nitrate will not be drew down to a low level by denitrification and anammox should be discussed.
- If possible, it will be better to details sampling and preservation of anaerobic waters, cause samples may undergo oxidation during sampling and preservation.
- It will be much better to draw the vertical profile of oxygen, temperature and density in the paper.
- Line 14:”water boy” should be “water body”.
Author Response
see attachement

Reviewer 2 Report
The MS is of interest and fix to the general scope of the journal. However, some issues must be addressed by the authors. Comments have been embedded through the MS in order to help the authors to improve the current version. The main concern is related to the monitoring of nitrogen compounds. Nitrogenous gasses have not been monitored, consequently, the work does not display a complete picture.

Author Response
see attachement

Round 2
Reviewer 2 Report
Thank you very much for addresing all the comments made by this reviewer.